# Air Change in Low and High-Rise Apartments

**Yonghang Lai [1], Ian A. Ridley [1] and Peter Brimblecombe [1,2,3,4,*]**

1   School of Energy and Environment, City University of Hong Kong, Hong Kong;
    yonghalai2-c@my.cityu.edu.hk (Y.L.); ian.alex.ridley@gmail.com (I.A.R.)
2   Department of Marine Environment and Engineering, National Sun Yat-Sen University,
    Kaohsiung 80424, Taiwan
3   Aerosol Science Research Center, National Sun Yat-Sen University, Kaohsiung 80424, Taiwan
4   School of Environmental Sciences, University of East Anglia, Norwich NR4 7TJ, UK
*   Correspondence: p.brimblecombe@uea.ac.uk

**Abstract:** Air exchange in tall apartment buildings is critical in controlling indoor environments in urban settings. Airtightness is relevant to energy efficiency, thermal comfort and air quality experienced by urban dwellers who spend much of their time indoors. While many air change measurements have been made in residential homes, fewer are available for high-rise apartments. The blower-door and $CO_2$ exchange methods were used to measure air change in some Hong Kong apartment buildings, for comparison with those from other parts of the world. Hong Kong apartments are often small and typical rented apartments show a median of seven air changes per hour under a 50 Pa pressure difference, similar to Mediterranean houses, though much greater than the airtight buildings of Northern Europe. Extrapolation of blower-door measurements made at 50 Pa to the natural pressure difference measured for individual Hong Kong apartments provides an approximation (within 8%) of the natural air change rate measured with a tracer. Air flow is a function of the pressure difference $\Delta P^{n_f}$ and the exponent $n$ was found close to the typical 0.6. There was a positive relationship between air permeability and construction age, but some of this also seems to reflect varying levels of maintenance by the building management companies. The median exchange in the apartments under naturally ventilated conditions was $0.26 \ h^{-1}$, not atypical of some houses on the US West Coast.

**Keywords:** airtightness; blower-door; $CO_2$ exchange effective leakage area; Hong Kong; indoor climate; indoor air pollution; permeability

## 1. Introduction

Tall apartment buildings are increasingly a characteristic of high-density urban living and thus represent an environment where those dwelling in cities may spend a great deal of time. Air change is a key parameter in describing the exchange of room air with the ambient environment, so it affects indoor air pollution, yet is also relevant to other issues, such as thermal comfort and energy efficiency. The exchange of air is often expressed as the air change rate, i.e., the rate at which air is replaced. Conventionally, this is expressed as the number of air changes per hour (ACH), it is affected by factors such as the design of the building and occupant behavior, weather variation and building ventilation mode. There are three main ventilation modes [1]: (i) mechanical ventilation, (ii) natural ventilation driven wind or buoyancy-induced flow through open windows or doors, and (iii) infiltration through gaps and cracks in the building envelope. Unlike other mechanisms, infiltration cannot be controlled, as it represents the ventilation mode when all windows and doors are closed.

In the Hong Kong Special Administrative Region, most residents will close all windows and doors during periods when air-conditioning is required, and during cold periods to prevent drafts. Airtightness thus becomes an important factor in building efficiency [2] and affects exposure to

pollutants indoors [3]. There are few surveys of airtightness tests or comprehensive data sets for the region, and little in terms of standards for air change. Concern about airtightness has persisted for some decades [4,5], especially as Hong Kong is such a crowded city, and Guidance Notes now set ventilation rates for some building types [6]. In contrast, air change in residential dwellings has been well-studied in Europe and North America (some examples in Table 1), although much research has focused on houses, with less work on the tall residential buildings that characterize the cities of Asia. In this paper we explore the air change rate in some Hong Kong apartments and compare these with other cities, aiming to establish a context for our measurements in apartment buildings and their relevance to regulating the built environment. Houses are very rare in Hong Kong and are typically in rural areas, so they have not been studied.

**Table 1.** Previous work used.

| Country | Building | $k_{50}$/h$^{-1}$ | $p$/m s$^{-1}$ | $A_{ELA}$/cm$^2$ | Reference |
|---|---|---|---|---|---|
| Korea | High-rise | $\bar{x}, \sigma_{Sn}$ <br> 3 ± 0.5 | | | Shin and Jo [7] |
| China | Low-rise | $\widetilde{x}$, iqr <br> 6.5, 5.65 | | $\widetilde{x}$, iqr <br> 189,174 | Chen et al. [8] |
| Russia | High-rise | $\widetilde{x}$, iqr <br> 7.5, 3.5 | | $\widetilde{x}$, iqr <br> 232, 143 | Armstrong et al. [9] |
| US | Public | | $\bar{x}, \sigma$ <br> 7.3 ± 0.2 | | Bahnfleth et al. [10] |
| Canada | Houses | $\bar{x}, \sigma_{Sn}$ <br> 3.1±1 | | | Hamlin & Gusdorf [11] |
| Estonia | Houses | $\bar{x}, \sigma$ <br> 4.9 ± 3.5 | $\bar{x}, \sigma$ <br> 4.3 ± 3.5 | | Kalamees [12] |
| Finland | Houses | $\bar{x}, \sigma$ <br> 3.7 ± 2.2 | | | Jokisalo et al. [13] |
| Denmark | Houses | - | | | Tommerup et al. [14] |
| Norway | Houses | $\bar{x}, \sigma_{Sn}$ <br> 3.9±1 | | | Granum & Haugen [15] |
| Italy | Houses | $\widetilde{x}$, iqr <br> 6.6, 1.9 | $\widetilde{x}$, iqr <br> 5.5, 2.8 | | Alfano et al. [16] |
| Greece | Houses | $\widetilde{x}$, iqr <br> 7.6, 4.2 | | | Sfakianaki et al. [17] |
| Ireland | Houses | $\widetilde{x}$, iqr <br> 10, 5 | $\widetilde{x}$, iqr <br> 9.7, 4.7 | | Sinnott and Dyer [18] |
| UK | Houses | | $\widetilde{x}$, iqr <br> 13, 5 | | Johnston et al. [19] |
| UK | Low-rise | | $\bar{x}, \sigma_{Sn}$ <br> 5.3 ± 0.3 | | Pan [20] |
| UK | Houses | | $\bar{x}, \sigma_{Sn}$ <br> 11 ± 8 | | Stephen [21] |
| UK | Houses | | $\bar{x}, \sigma$ <br> 18 ± 7 | | Hong et al. [22] |
| UK | Houses | | $\bar{x}, \sigma_{Sn}$ <br> 9 ± 3 | | Grigg [23] |

Notes: $k_{50}$ air change rate at 50 Pa pressure difference, $p$ is permeability and $A_{ELA}$ is the effective leakage area with central tendency as median ($\widetilde{x}$,) or mean ($\bar{x}$), and dispersion as interquartile range (iqr), standard deviation ($\sigma$) or estimated from Snedecor's rule ($\sigma_{Sn}$) or small (-). Note: Korean building values represent an average of four and the Canadian work only includes values of leakier building.

## 2. Materials and Methods

### 2.1. Data Sources

This paper relies on data sets for air change mostly in residential dwellings (Table 1), in addition to using a small set from our own measurements made from apartments in Hong Kong (Figure 1). Thousands of fan-pressurization measurements (i.e., blower-door tests), usually made at a 50 Pa pressure difference, have been conducted in US buildings [24] and relate air exchange to the type of construction, region, the number of stories, floor or basement type and age of dwellings. Previous studies, along with those shown in Table 1, establish the role of design and construction, building supervision and the quality of workmanship, which may sometimes be more important than building age or resident and management behavior e.g., [8,18]. The junction of ceilings/floors with external walls, the junction of separating walls with the external wall, gaps between electrical and plumbing installations or chimney and ventilation ducts, and gaps and cracks between windows and doors are all important places for leakage. In some countries, especially those in Northern Europe, airtightness is well-controlled, but elsewhere it may be lax. China may not meet international standards, so increasing airtightness could reduce residential energy use by 12.6% [8].

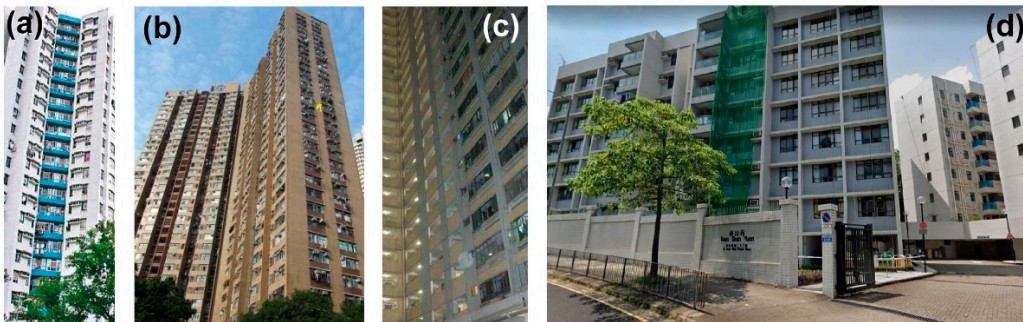

**Figure 1.** Photographs of some Hong Kong buildings that formed part of the study for (**a**) Site 4; (**b**) Site 5; (**c**) Site 6; and (**d**) Site 7. No photograph of the building with apartment sites 1–3, 8–10.

### 2.2. Study Apartments in Hong Kong

The current study adds measurements of air change at 50 Pa from 10 apartments located in Hong Kong. The region has a warm-humid subtropical climate, with an average temperature above 30 °C for the four warmest months, while winter months rarely drop to 10 °C. Hong Kong has more than seven million inhabitants crowded into limited areas of flat land. Most people in the city tend to live in small apartments in high-rise buildings due to the lack of flat land, which leads to some of the highest population densities in the world (59,000/km$^2$ in Kwun Tong). The city is the world's tallest agglomeration, with most inhabitants living on the 15th floor or higher. Crowding is such that some dwellings are very small, so-called coffin homes [25] of some 12 m$^2$, where poor ventilation (i.e., low air change rates) may be a problem [5,26]. Ironically, low ventilation rates are an issue which these very substandard dwellings share with highly efficient houses, for example in Sweden, where air change can fall below 0.3 h$^{-1}$ [27]. The few measurements available in Hong Kong examine air ventilation rates in high-rise residences with air conditioning [28]. These suggest that out of 15 apartments, 12 tested at high fan speeds showed a median air change rate of 4.9 h$^{-1}$ (lower quartile: 4.5 h$^{-1}$; upper quartile: 5.15 h$^{-1}$).

The indoor measurements of air change made in the present study examine apartments spread among five buildings located in the New Territories of Hong Kong (Table 2). The buildings were built between the 1980s and 1990s, with construction age estimated from the first year of occupancy. The envelope area was determined from all the physical separators (walls, floors, ceilings, etc.) that demark indoors and outdoors. The total length of significant penetration paths was determined as the length of some obvious gaps or cracks, including those associated with windows and doors. It is

important to note that Hong Kong living spaces are often small; in this study high-rise apartments had areas of $41 \pm 16$ m$^2$ and low-rise $86 \pm 41$ m$^2$. All the low-rise apartments (in buildings of ten floors or fewer) had balconies, and while balconies are present in some high-rise buildings in Hong Kong, these are not for the faint of heart. These can act as sun-shades and can mitigate air pollution and traffic noise [29], though the buildings in this study had no balconies.

**Table 2.** Apartments in high-rise and low-rise buildings studied in the Kowloon and Shatin districts of Hong Kong with the number of floors. The site number, floor of apartment floor area ($A_F$) and volume ($V$), along with the outdoor ($T_o$) and indoor ($T_i$) temperature, wind speed ($u$) and the natural indoor-outdoor pressure difference ($P_{nat}$). See Figure 1 for photographs of buildings in this study: 1(a) Sites 4 and 1; (b) Sites 5 and 1; (c) Sites 6 and 1; (d) Site 7. No photographs are available for Sites 1–3 or 8–10.

| Building | Site | Floor | $A_F$/m$^2$ | $V$/m$^3$ | $T_o$/ °C | $T_i$/ °C | $u$/km h$^{-1}$ | $P_{nat}$/Pa |
|---|---|---|---|---|---|---|---|---|
| | 1 | 12 | 38.9 | 101.3 | 27 | 29 | 5.0 | |
| High-Rise 34 | 2 | 16 | 53.5 | 139.1 | 26 | 24 | 6.6 | |
| | 3 | 16 | 53.5 | 139.1 | 27 | 28 | 8.6 | 0.62 |
| High-Rise 28 | 4 | 11 | 54.8 | 142.4 | 27 | 26 | 5.0 | 0.74 |
| High-Rise 30 | 5 | 30 | 26.7 | 69.4 | 30 | 31 | 11.9 | |
| High-Rise 23 | 6 | 19 | 16.5 | 42.9 | 29 | 32 | 4.3 | |
| Low-Rise 5 | 7 | 5 | 146.8 | 381.7 | 27 | 27 | 8.0 | 0.35 |
| | 8 | 6 | 61.9 | 160.9 | 29 | 32 | 3.9 | 0.14 |
| Low-Rise 8 | 9 | 6 | 61.9 | 160.9 | 30 | 32 | 3.9 | 0.12 |
| | 10 | 4 | 71.7 | 186.4 | 28 | 30 | 3.9 | 0.21 |

All buildings examined here are brick and concrete structures with no heat conservation materials in the walls. The floor structure of all buildings is floor slab. All internal walls between different units are drywall. Compared with concrete walls, airtight joints are not easily constructed, so a lower airtightness level was expected due to these internal walls. All external walls are concrete and have punched windows. Typically, the apartments have casement windows with steel frames, except for a large awning window at Site 7. Windows represent places where air can leak into the apartments.

*2.3. Measuring Air Change Rate and Effective Leakage Area*

Measurements of airtightness in buildings are often expressed as the air change rate, which typically takes the unit h$^{-1}$. It can be measured with tracers, but often uses a blower-door, which is readily available equipment and adopts procedures (following ISO 9972) that are well-developed [30]. Such fan-driven airtightness tests can also be used to identify the air flow rate required to determine particle deposition rates and penetration factors [31]. The method is useful because the effects of mechanical ventilation systems (often known as heating-ventilation-air-conditioning, HVAC) and fluctuating natural ventilation (window opening) can be greatly reduced.

Fan pressurization sets a steady-state flow to maintain a steady pressure across the building envelope. Here we used a typical Retrotec Blower Door System (Retrotec, Everson, WA USA) consisting of:

1.  a door panel which temporarily seals a typical doorway and provides a large circular hole to mount a fan;
2.  a calibrated fan capable of creating a measurable flow of air;
3.  a pressure gauge that can calculate air flow through the fan.

The fan pressure is the pressure along the axis of the fan, and it allows air flow to be calculated. Pressure differences between two locations, rather than absolute pressures, were measured using digital manometers, which show little drift during the hour-long tests. Measurements are made both under pressurization and depressurization, which may reveal differences commonly attributed to

physical changes in the pressure boundaries of the envelope, e.g., depressurization may pull a window more tightly shut. The air flow rate is often reported at a single pressure point, conventionally 50 Pa, e.g., [20,32,33]. This is high enough to overcome pressure noise and drift caused by wind or stack effects [34], but, unfortunately, the flow at 50 Pa cannot represent the air flow rate under natural conditions. However, the power-law function allows the relationship between air flow and pressure difference to be expressed as:

$$Q_f = C\,\Delta P_\mathrm{f}{}^n \tag{1}$$

where $Q_f$ is the air flow (m$^3$ h$^{-1}$), $C$ is the flow coefficient (m$^3$ h$^{-1}$ Pa-n), $\Delta P$ is the indoor-outdoor pressure difference (Pa), $n$ is the pressure exponent and the subscript f relates to the fan-induced flow or pressure, with $C$ and $n$ determined by least squares fitting. Air change rate ($k_{ACH,\Delta P}$) aims to measure how frequently the air within an interior space is replaced, typically as air changes per hour at a given pressure:

$$k_{ACH,\Delta P} = Q_{f,\Delta P}/V \tag{2}$$

where $V$ is the volume. We can express the movement into the room as permeability ($p$):

$$p = Q_{f,\,\Delta P}/A \tag{3}$$

which can be considered the flow through a given area of the building envelope (i.e., a flux), and in the case of a volume of air moving into a building it has the units m h$^{-1}$ (although often written m$^3$ h$^{-1}$/m$^2$), so it can be thought of as a permeation velocity. The effective leakage area ($A_{ELA}$/m$^2$) was calculated following the procedure in ASTM E779-10 [35]. A discharge coefficient of 1.0 and a reference pressure of 4 Pa are used for calculating the ELA, as shown in Equation (4):

$$A_{ELA} = C\Delta P_{4}{}^{n^{-1/2}}\,(\rho/2)^{1/2} \tag{4}$$

where $\Delta P4$ is the reference pressure of 4 Pa, and $\rho$ is the density of the air (kg m$^{-3}$).

The *FanTestic* software, associated with the blower-door equipment, utilizes building characteristics (e.g., volume, envelope area, floor area, apartment height) and temperatures. Experimental runs during the blower-door tests combined ten or more measurements of the flow rate to achieve stable values at each pressure point over the range 10–60 Pa. This overcomes the effects of swings caused by changing wind or elevator movements. Periods of excessive measurement uncertainty are detected by the software, so the measurement can be repeated until an acceptable level of uncertainty is achieved.

## 2.4. Measuring Natural Ventilation Rate

The tracer gas technique is also commonly applied (adopting ISO 20485) to determine the air change rate [36,37]. It involves injecting a tracer gas and mixing it through a room, then calculating its decay rate from concentration profiles. In this study, $CO_2$ was used as a tracer gas to detect the air change rate under natural ventilation (i.e., $k_{CO_2}$) using HOBO $CO_2$ loggers (HOBO MX1102, Onset Inc., USA). At the very start of the decay tests, the loggers were placed in different corners of the room and confirmed that the air was well-mixed. Some incense sticks were burnt to raise $CO_2$ to exceed 1000 ppm; the source was then removed. During the test, all windows and external doors were closed, so the natural ventilation rate was estimated under the minimum ventilation mode. The measurements were made at midnight, a time when external sources were at a minimum, so ambient $CO_2$ concentrations are almost constant and wind speeds low, such that the effects of wind on air change were kept to a minimum, as these can cause variability in tall buildings. The decay curve was used to calculate the rate, assuming infiltration/exfiltration rates of the tracer gas are constant in the well-mixed space and there is no chemical transformation:

$$k_{CO_2} = \ln(f_0)\{(c - c_{out})/(c_M - c_{out})\}/(t - t_0) \tag{5}$$

where $kCO_2$ is the air exchange rate ($h^{-1}$), and $c$ and $c_M$ are the tracer gas concentrations (ppm) measured at $t$ and $t_0$, the end and beginning of the decay curve, and $c_{out}$ is the outdoor concentration at time $t$.

### 2.5. Statistical Analysis

Two measures of central tendency and dispersion were used in this study, most often the median ($\widetilde{x}$) and upper ($Q_3$) and lower quartiles ($Q_1$); these measures were adopted because the data often has a non-normal distribution and a positive skew. In some cases, the original data was not accessible and was reported as the mean ($\overline{x}$) and standard deviation ($\sigma$). In data sets where only the average and the range were available, Snedecor's rule [38] was used to estimate the standard deviation ($\sigma_{Sn}$). The medians and quartiles were calculated using Microsoft Excel, which makes linear interpolations where exact values are absent. Bivariate analysis used online tools from Wessa (https://wessa.net/) to determine the non-parametric Kendall rank correlation coefficient ($\tau$), rather than Pearson's $r$. The Mann–Whitney test (statistic $U$), a non-parametric equivalent of the $t$-test, was used to compare sample sets. Where there were more than two sets of data, one-way analysis of variance used the Kruskall–Wallis test. A non-parametric equivalent of ANOVA, and the Wilcoxon signed-rank test (statistic $W$), a non-parametric test, allowed two matched samples (equivalent to the paired $t$-test, also used in this study) to be compared. These tests were carried out using the online software Vassarstat (http://vassarstats.net/).

## 3. Results and Discussion

### 3.1. Air Change Rate in Apartments

The air change rates at 50 Pa determined for the Hong Kong apartments under both pressurization and depressurization are listed in Table 3. At 50 Pa, the air change rate lies in the range of 4–27 $h^{-1}$, with a median of 7.16 $h^{-1}$ ($Q_1$ = 5.2 $h^{-1}$; $Q_3$ = 13.2 $h^{-1}$) for the high-rise apartments, and for the low-rise apartments a median of 7.32 $h^{-1}$ ($Q_1$ = 6.5 $h^{-1}$; $Q_3$ = 8.0 $h^{-1}$). The values measured in Hong Kong are compared with those elsewhere in Figure 2. As noted in Table 1, data are summarized in terms of the median with dispersion as the interquartile range, and where this was not possible it is expressed as the mean and standard deviation.

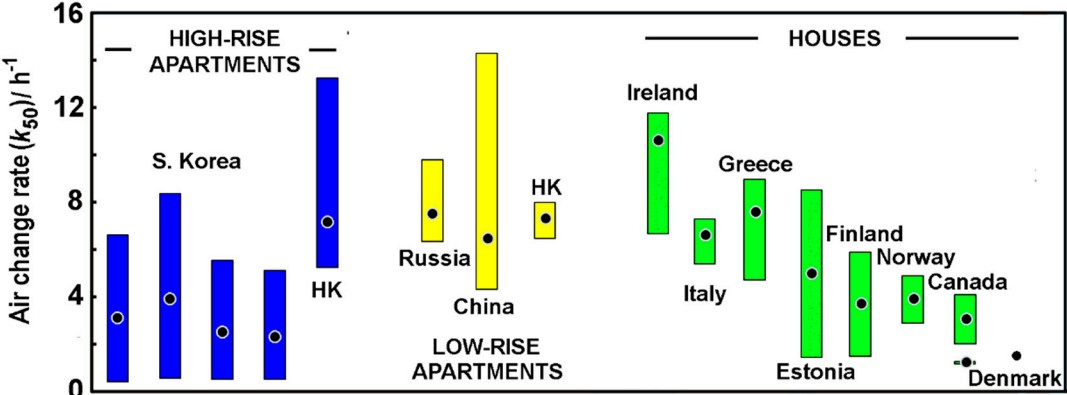

**Figure 2.** Air change rates at 50 Pa for apartments in high- and low-rise buildings and detached houses from the studies listed in Table 1. Note: where possible, these are plotted as bars which represent the interquartile range, and where this is not available, it is the mean and standard deviation calculated conventionally or using Snedecor's rule, as noted in Table 1. Black dots denote the median or mean.

**Table 3.** Air change ($k_{50}$) rate, the permeability ($p$) and effective leakage area ($A_{\mathrm{ELA}}$) for the Hong Kong apartments under both depressurization and pressurization at a pressure difference of 50 Pa. *Note: the percentage difference is determined as $200\,(k_{50,D} - k_{50,P})/(k_{50,D} + k_{50,P})$.

| Site | Test Mode | $k_{\mathrm{ACH}}$ $h^{-1}$ | Difference * % | $p$ $m\ h^{-1}$ | $A_{\mathrm{ELA}}$ $cm^2$ |
|---|---|---|---|---|---|
| 1 | Depressurization | 6.50 | 10.0 | 4.81 | 201 |
|   | Pressurization | 5.88 |  | 4.35 | 181 |
| 2 | Depressurization | 7.90 | 1.2 | 5.40 | 343 |
|   | Pressurization | 7.61 |  | 5.34 | 330 |
| 3 | Depressurization | 5.27 | 2.1 | 3.93 | 224 |
|   | Pressurization | 5.16 |  | 3.85 | 219 |
| 4 | Depressurization | 4.61 | 10.0 | 3.44 | 195 |
|   | Pressurization | 4.17 |  | 3.11 | 177 |
| 5 | Depressurization | 13.18 | −1.1 | 11.61 | 279 |
|   | Pressurization | 13.38 |  | 11.78 | 283 |
| 6 | Depressurization | 26.71 | 4.0 | 20.21 | 349 |
|   | Pressurization | 25.67 |  | 19.42 | 336 |
| 7 | Depressurization | 4.32 | −15.6 | 4.03 | 503 |
|   | Pressurization | 5.05 |  | 4.72 | 588 |
| 8 | Depressurization | 7.33 | −10.5 | 6.16 | 359 |
|   | Pressurization | 8.14 |  | 6.84 | 399 |
| 9 | Depressurization | 7.31 | −11.1 | 6.14 | 359 |
|   | Pressurization | 8.17 |  | 6.86 | 401 |
| 10 | Depressurization | 6.94 | −13.4 | 6.13 | 394 |
|   | Pressurization | 7.94 |  | 7.01 | 451 |

The high-rise apartments in Hong Kong seem to have greater air exchange rates than those in South Korea. The 350 dwellings examined by Shin and Jo [7] were from four modern high-rise buildings in Incheon. They showed that the average air change rate at 50 Pa was 2.9 $h^{-1}$, so these modern buildings are quite tight when expressed as an ASHRAE level, which suggests better performance than found in the older and more crowded buildings tested in Hong Kong. Here some windows cannot be kept tightly closed (notably at Site 6). The low-rise buildings of Northern China, Russia (listed in Table 1) and Hong Kong are some 20–40 years old and show high air change rates. The Chinese buildings in Beijing and Tangshan varied greatly, with one apartment showing an air change rate of 27 $h^{-1}$. Chen et al. [8] describe the "terrible air tightness performance" and compared it with the Chinese standard JGJ26-95 of 8.5 $h^{-1}$. In these apartments, the inadequate performance seemed to arise from poor window closure and a lack of occupant awareness. Houses tended to perform better in colder countries (Canada, Denmark, Estonia, Finland and Norway), while houses in Mediterranean Italy and Greece were much leakier. Ireland exhibited rather leaky homes, but also included some older buildings with single glazed windows in good condition; these in many cases had been painted over, such that they could not open and the buildings were, therefore, more airtight [18]. The UK and the US often showed examples of poor performance e.g., [21], but this could depend on housing condition; the air change rate in low-income row houses in Philadelphia was ~60 $h^{-1}$ [39], while multifamily buildings in Upstate New York prior to retrofit had an air change rate of ~35 $h^{-1}$ [40].

A relatively large variation is found between the air change rate at 50 Pa of depressurization and pressurization for the high-rise apartments, although, in general, depressurization leads to larger air exchange rates. In contrast, in the low-rise apartments depressurization leads to lower air exchange rates, with the largest difference being about −16% at Site 7. A similar relationship was also observed in seven dwellings in the UK [19], where pressurization led to significantly larger air exchange ($W = -26$; $p_1 \sim 0.02$; $N = 7$) in all but one dwelling. Four of the five low-rise apartments in Northern China [8]

show the reverse, with pressurization leading to more rapid air change, although the small sample size makes the statistical certainty low. These differences in response under depressurization and pressurization can suggest different approaches to construction or maintenance of the integrity of inner and outer wall layers of the buildings. Under pressurization, greater indoor pressure means that the inner wall layer will be pushed outward so some points may leak more, such as fan louvres; while during depressurization air is sucked inside so some outer wall layers may bend inwards and leak. The values measured here can be compared with the standards tabulated by Pan [20] which are typically 2.8–6 $h^{-1}$ for most European countries, although for Finland, which has harsh winters, it has been set to 1 $h^{-1}$.

## 3.2. Changing Pressure and Natural Ventilation Rates

The measurements made in the Hong Kong apartments took place over a range of pressures so it was possible to determine the air flow $Q_f$ over a range of pressures (Figure 3) that are described in terms of Equation (1), along with $n$ as the pressure exponent. The software associated with the Retrotec 6100 Blower Door System calculates this via least squares. This exponent would be expected to vary between 0.5 (turbulent) and 1.0 (laminar) but, as flow into buildings is through orifices, values closer to 0.5 are to be expected. In this work, $n$ was found to be 0.59 ± 0.05. A paired t-test suggested that the exponent was slightly, though significantly ($p_1 < 0.06$), smaller under depressurization (0.57 ± 0.01 compared with 0.61 ± 0.03). This is close to the suggestion in Numerical Data for Air Infiltration and Natural Ventilation Calculations [41] that this exponent is typically 0.60, but with a range of ±0.1. Alfano et al. [16] found $n$ to be 0.60 ± 0.035 for 20 Italian houses, while the three classes of Irish houses (pre-1975, 1980s and ~2008) had $n = 0.64 ± 0.018$ [18], although an ANOVA test suggested no significant difference among them. In theory, Equation (1) could be used to extrapolate from 50 Pa to more realistic pressure differences, but wind and stack effects in taller buildings are likely to make this unreliable.

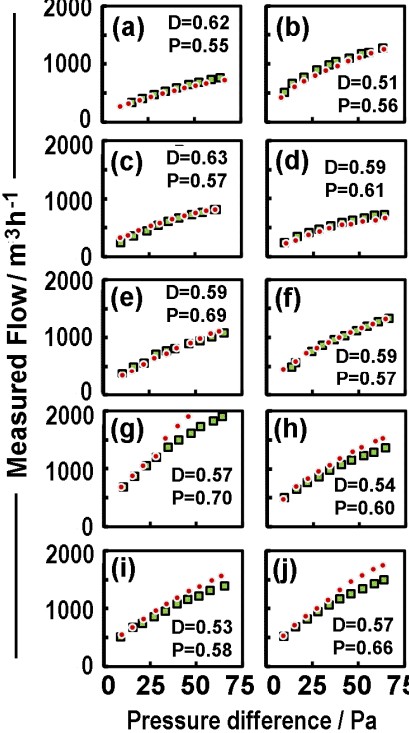

**Figure 3.** Air change rate as a function of the pressure difference for the 10 study apartments in order of Sites 1–10. Note: depressurization (D) is denoted by squares and pressurization (P) by dots, and the exponent *n* for both conditions is marked on each of the 10 panes.

The natural ventilation rates were measured using carbon dioxide as a tracer in two of the high-rise and four of the low-rise apartments in Hong Kong. Sets of three measurements were made with carbon dioxide at each apartment, and the median exchange rate for the six apartments was 0.26 h$^{-1}$ ($Q_1$=0.18 h$^{-1}$; $Q_3$=0.31 h$^{-1}$), which is about an order of magnitude less than at 50 Pa. The natural air change rates are only slightly greater than measurements from detached houses in the Pacific North-West: $\tilde{x}$ = 0.19 h$^{-1}$; $Q_1$=0.17 h$^{-1}$; $Q_3$=0.23 h$^{-1}$ [42] and 0.14–0.22 h$^{-1}$ [43], but are much less than low-income apartments in New York City: 1.08 h$^{-1}$, 0.58 h$^{-1}$ and 1.01 h$^{-1}$ [44], or a 46-year-old house in Indiana: $\tilde{x}$ = 0.68 h$^{-1}$; $Q_1$=0.5 h$^{-1}$; $Q_3$=0.82 h$^{-1}$ and its newer companion, which performed better: $\tilde{x}$ = 0.32 h$^{-1}$; $Q_1$=0.315 h$^{-1}$; $Q_3$=0.36 h$^{-1}$ [45].

Typically, measurements of air change are made at 50 Pa, but these can be difficult to translate to the natural air change rate measured using a tracer such as the $CO_2$ used here. Sherman and co-workers [24] suggest that the air change rate under natural conditions, estimated with a tracer such as $CO_2$, is typically $k_{50}/20$, the factor of 20 is derived from fits to a large amount of data from houses in the US. In the current study, the values for this factor ranged between 13 and 30, as suggested in Figure 4a, by the 1:20 line. The high variability shown in this factor was not surprising given the complexities of stack and wind effects in tall buildings; besides this, our sample size is small (N = 6). Equation (1) allows blower-door measurements of air change at higher pressures to be extrapolated to the natural pressure differences measured in the study apartments (listed in Table 2). These are displayed in Figure 4b and while the calculated values of air change rate are a little higher than those measured with the $CO_2$ tracer, they are reasonably close to the 1:1 line.

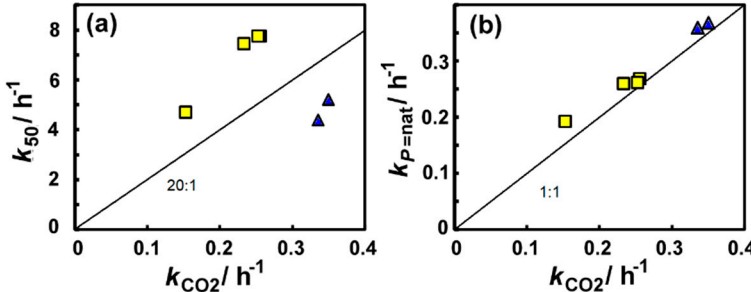

**Figure 4.** Natural air change rates (**a**) Comparison between the air change rate measured with the $CO_2$ tracer and the blower-door at 50 Pa. The line represents the 20:1 slope expected for US houses [24]; (**b**) Comparison between the air change rate measured with the $CO_2$ tracer and blower-door measurements at 50 Pa extrapolated to the natural pressure difference of each apartment (Equation (1)). The line represents the 1:1 slope for perfect agreement. Note: high-rise apartments: dark triangles; low-rise apartments: squares.

Small low-income apartments in Hong Kong are poorly ventilated, and the carbon dioxide concentration profiles displayed in Cheung and Jim [5] suggest relaxation times of around an hour. This is much like the natural air change rates within an hour found in low-income apartment buildings in New York. Although there is much concern that the air exchange is too low, it is greater than that of Hong Kong low-rise apartments occupied by professionals, where the median natural exchange rate is 0.19 h$^{-1}$. However, Hong Kong's coffin homes are tiny, with floor areas of around 10 m$^2$ and total volumes of ~35 m$^3$, so the actual volume of air exchanged may be very low. Additionally, better quality apartments have effective vents in the kitchens, to remove moisture and pollutants created during cooking. While there is much desire to achieve lower air exchange rates from a point of energy efficiency, ventilation can become reduced to excess. Swedish regulations require at least 0.5 h$^{-1}$, but more than a third of single-family houses there were found to have air change rates lower than 0.3 h$^{-1}$ [27]. In this context, the Hong Kong results are not surprising. A balance of competing factors has to be considered when expressing the airtightness of buildings, whereby natural ventilation rates of 1.5 h$^{-1}$ are said to be indicative of low airtightness, 0.8 h$^{-1}$ of medium airtightness, and 0.5 h$^{-1}$ reaching a high level

of airtightness [17]. However, when such air change rates fall this low (i.e., 0.5 h$^{-1}$), it is likely to contravene the ventilation requirement of some national building regulations. Persily [46] traces the history of this delicate balance between the provision of outdoor air to reduce the concentration of internally generated contaminants and the desire for decreased ventilation since the 1980s; this has been motivated by a desire to reduce energy consumption within buildings.

### 3.3. Permeability and Leakage Areas in Apartments

Table 3 also lists the air permeability at 50 Pa for the Hong Kong test apartments. It suggests that around 80% of the apartments studied achieved the air permeability standard in Part L1a 2006, i.e., 10 m h$^{-1}$, and the majority (75%) achieved the Part L1a Indicative Standard for the Standard Assessment Procedure 2005 of 7 m h$^{-1}$ [47]. Only 40% of the test apartments were lower than 5 m h$^{-1}$, which would comply with the EST Good Practice, but none of apartments meet Best Practice at 3 m h$^{-1}$ [48–50]. However, the permeability of Hong Kong apartments is not dissimilar to those of Northern Europe and some North American low-rise public buildings (Figure 5). The measured values can also be compared with guideline values of 3 m h$^{-1}$ for Estonia and 1 m h$^{-1}$ for Canada, as tabulated by Pan [20].

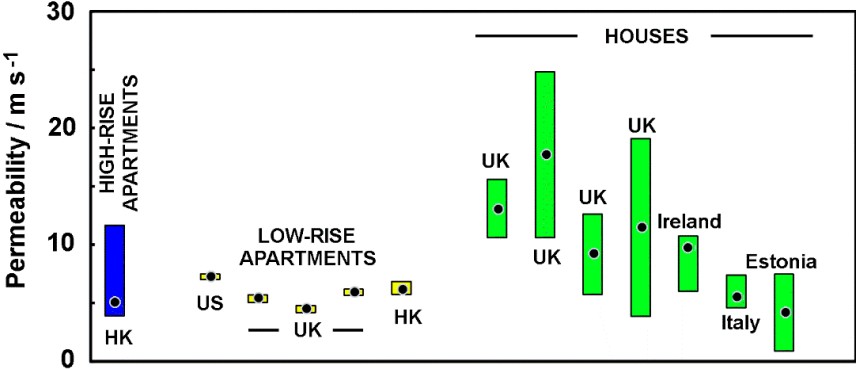

**Figure 5.** Permeability for apartments in high- and low-rise buildings and detached houses from the studies listed in Table 1. Note: where possible, these are plotted as bars which represent the interquartile range and where this is not available, it is the mean and standard deviation calculated conventionally or using Snedecor's rule, as noted in Table 1. The black dots denote the median or mean.

Additionally, Table 3 lists the Effective Leakage Area ($A_{ELA}$) in the Hong Kong test apartments ($\widetilde{x}$ = 340 cm$^2$; $Q_1$ = 223 cm$^2$; $Q_3$ = 395 cm$^2$), and not surprisingly this is significantly greater in the higher quality, though larger, low-rise buildings ($\widetilde{x}_{LR}$ = 360 cm$^2$; $\widetilde{x}_{HR}$ = 251 cm$^2$; Mann-Whitney $U$ = 96, $p_1$ < 0.0002). Leakage can also be expressed as a ratio that allows for building size, as the normalized leakage area (i.e., $A_{ELA}/A_{Envelope}$). The LEED standard of the US Green Building Council suggests residential units must "demonstrate less than 1.25 square inch leakage area per 100 square feet of enclosure area."—this is a ratio of 8.7 × 10$^{-5}$ and alternatively takes a value of 7 × 10$^{-5}$ under the Canadian Standard R-2000 [51]. Neither of these standards are met by the Hong Kong apartments ($\widetilde{x}$ = 17.5 × 10$^{-5}$; $Q_1$ = 13 × 10$^{-5}$; $Q_3$ = 21 × 10$^{-5}$), although the "terrible air-tightness" found in some apartment buildings of Northern China [8] means that these (i.e., $\widetilde{x}$ = 25 × 10$^{-5}$; $Q_1$ = 20 × 10$^{-5}$; $Q_3$ = 66 × 10$^{-5}$) also fail the two North American standards.

### 3.4. Airtightness and Building Characteristics

Figure 6a shows the relationship between air permeability and age, i.e., years since first occupation. It reveals a significantly positive relationship using the Kendall rank correlation ($\tau$ = 0.69; $p_2$ < 0.0002; $N$ = 20). This shows that older apartments have a higher air permeability, in line with many earlier studies, although some (e.g., [18]) suggest that new dwellings cannot automatically be assumed to be

more airtight than older dwellings. As seen in Figure 6b, the large apartments, in terms of floor area, had lower permeability, possibly arising from generally more effective maintenance.

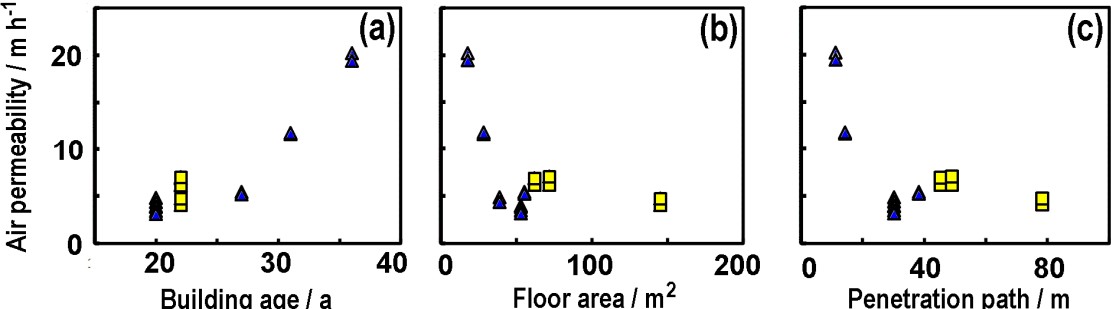

**Figure 6.** Permeability for apartments in Hong Kong as a function of (**a**) age; (**b**) floor area AF; and (**c**) measured length of the penetration path. The high-rise apartments are marked as triangles and the low-rise ones as squares.

Pan [20] reported a weak positive correlation between measured air permeability and the number of significant penetration paths through the dwelling envelope. The penetration path here refers only to the gaps between window frames and walls, doors and walls, and cracks between extractor fans or air conditioners and walls, taken to represent the obvious routes for air leakage. Figure 6c shows the relationship between air permeability and the total penetration path; convincing visually, though not statistically ($p_2 > 0.5$). However, the relationship is driven by some smaller apartments with limited window areas, which despite this are quite permeable, hence the apparent disagreement with Pan [20]. In Hong Kong, larger apartments are often found in low-rise buildings and are occupied by wealthier families. These may be better designed and constructed, with a higher level of maintenance. In contrast, some apartments in high-rise buildings are older and poorly maintained. Observations from Site 5 revealed that the dropped ceilings and the window frames were poorly fitted and had noticeable gaps. As this apartment was located atop a 30-floor high-rise building, greater wind speeds could further enhance air permeability. At Site 6, some windows could not be fully closed, yet there was little enthusiasm for repair.

### 3.5. Methodological Issues or Air Balance in High-Rise Apartments

Stack effects are especially noticeable in high-rise buildings and cause elevator doors to stick, make it difficult to open doors to apartments and increase noise due to air leakage [52]. Typically, tall buildings can also create problems for the blower-door method, due to changes in pressure from the operation of elevators or large shifts in external wind speed. Our $CO_2$ tracer experiments had to be conducted around midnight, to gain advantage from low wind speeds and elevator use, but this made them more likely to be stable. During the current blower-door measurements, the issue of wind did not prove to be a problem when using a pressure difference of 50 Pa, but care was taken to establish a stable baseline pressure calibration from the start. Following this, the blower-door tests were only conducted while stability could be maintained, and subsequent measurements were repeated to reach an acceptable level of certainty. However, we became aware of issues related to the balance between infiltration air from outside compared to other internal neighboring spaces (apartments, corridors, elevator lobbies, etc.) when determining particle deposition as part of another study [31]. The proportion of the air which derives from the outside and that from other spaces within the building could not be accurately determined in the present study. It is possible to use an additional blower-door to isolate the neighboring or internal apartments or corridors to calculate the exchange of external air, as seen in the airtightness measurements from Korean high-rise buildings [7]. Such corrections were not possible in our study, as it would have required more equipment and permission to access other apartments. Thus, the air exchange parameters reported here relate to the overall airtightness rather

than one related only to external air. Some preliminary, yet unpublished modeling using CONTAM, suggested ~25% of the air arose from neighboring apartments and common areas in the buildings studied here.

## 4. Conclusions

Most people in Hong Kong live in small apartments in high-rise buildings, as the result of extremely high population density. Small apartments mean that there is a difficult balance between air exchange and a lack of protection from ambient air pollution and a loss of thermal efficiency. The blower-door test was used to measure the air change rate and air permeability, providing an easy way to determine airtightness for comparison with buildings from a variety of countries. It is clear that both high- and low-rise apartments in Hong Kong have relatively high air change rates at a 50 Pa pressure difference ($\tilde{x}$ = 7.3 h$^{-1}$; $Q_1$ = 5.2 h$^{-1}$; $Q_3$ = 8.1 h$^{-1}$), with high exchange even in the larger and well-maintained apartments occupied by professional people. Three quarters of these Hong Kong apartments met the UK Part L1a Indicative Standard for the Standard Assessment Procedure 2005 for air change rate, but most failed to achieve the normalized leakage areas under the LEED or R2000 requirements. Extrapolation of blower-door measurements made at 50 Pa to the natural pressure difference measured for individual apartments provides an approximation of the natural air change rate measured with a tracer. The estimates were, on average, about 8% greater than the measured value, though highly variable, with one 25% greater. Making better estimates is certainly something to be explored in further work.

A significant positive relationship between air permeability and the construction age is seen, but some of this also seems to reflect varying levels of maintenance by building management companies. The significant negative relationship between air permeability and total length of penetration path/envelope area is a characteristic of residential buildings, so it would ideally involve regular checks of sealing and obvious gaps, especially at the kitchen drain, windows and door frames. However, individual landlords, who may own just a few apartments in a given building, can see maintenance as little more than an added cost. In many ways, the rapid air change in Hong Kong apartments makes them akin to the houses of Mediterranean regions, which might seem reasonable given the city's warm climate, but the leakiness of the buildings imposes a load on air conditioning, which is required for many of the warmer months. Energy efficiency in buildings is important to Hong Kong, as air conditioning accounted for some 31% of the region's energy use, according to the *Hong Kong Energy End-use Data* 2019, published by its Electrical and Mechanical Services Department. There have long been pressures to reduce energy consumption with airtightness as a key approach. It may be possible to improve the energy efficiency through heat recovery units while maintaining the present ventilation level, but these units have not been widely applied. Future work needs to look at the role played by wind and stack effects in air exchange in high-rise apartments. Additionally, there needs to be more work on the way air moves between common areas and other apartments of tall buildings, as currently the balance of air deriving for outside and indoor spaces is not well understood.

Hong Kong sometimes experiences poor air quality on its busy streets, and aloft there is the potential for exposure to air pollutants transported down to the Pearl River Delta from an increasingly industrialized Guangdong Province. Airtight buildings can prevent the invasion of polluted air during air pollution episodes. At the other end of the scale, it is apparent that poorer people in Hong Kong living in tiny dwellings suffer from insufficient ventilation. In these dwellings, low air exchange may inhibit the removal of cooking fumes and carbon dioxide. The competing demands experienced in our cities are nicely reflected in the problem of ventilation described here: there are difficulties in establishing the right level of ventilation in the buildings of an increasingly urbanized world.

**Author Contributions:** Conceptualization of the investigation, methodology and supervision, I.A.R.; undertaking investigation, preparing analysis and original draft preparation, Y.L.; formal analysis and writing—review editing, P.B. All authors have read and agreed to the published version of the manuscript.

**Funding:** This work was supported by the Research Grants Council of the Hong Kong SAR under Grant number 9042397 (CityU 11334116).

**Acknowledgments:** We thank all the residents who provided their apartments during this work, particularly Frank Kit and Yao An.

**Conflicts of Interest:** The authors declare no conflict of interest.

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
