# Peer review of "Air Change in Low and High-Rise Apartments"

_urbansci, doi:10.3390/urbansci4020025_

Round 1

Reviewer 1 Report

Lai et al present an investigation of air exchange in low and high rise apartments in Hong Kong and the implications for their findings relative to previous studies and established building standards. They present concrete evidence for their findings in clear and concise manner. I recommend publication after a few minor corrections for grammar and clarity. Figures, tables, mathematical expressions are all appropriate, well formatted and insightful to the story told in the paper.

Minor revisions:

Line 49: "by contract" - I am uncertain what this is supposed to mean, please clarify.

Line 92: "These suggesting" - remove and start with Of fifteen apartments, or revise the sentence.

two un-numbered lines above line 272: super script missing on h^-1.

Line 294: awkward sentence structure (this motivated to reduce). Please revise.

Line 345 Section 3.5. This section suddenly starts using personal pronouns (we) should be revised to be consistent with the rest of the paper.

Author Response

REFEREE 1

Comments and Suggestions for Authors

Lai et al present an investigation of air exchange in low and high rise apartments in Hong Kong and the implications for their findings relative to previous studies and established building standards. They present concrete evidence for their findings in clear and concise manner. I recommend publication after a few minor corrections for grammar and clarity. Figures, tables, mathematical expressions are all appropriate, well formatted and insightful to the story told in the paper.

Minor revisions:

Line 49: "by contract" - I am uncertain what this is supposed to mean, please clarify.  DONE IT WAS "contrast"

Line 92: "These suggesting" - remove and start with Of fifteen apartments, or revise the sentence. REVISED "These suggest that of fifteen apartments, twelve tested at high fan speeds showed a median air change rate "

two un-numbered lines above line 272: super script missing on h^-1. DONE

Line 294: awkward sentence structure (this motivated to reduce). Please revise. CHANGED AS: "Persily [46] traces the history of this delicate balance between the provision of outdoor air to reduce the concentration of internally generated contaminants and the desire for decreased ventilation since the 1980s; this motivated by a desire to reduce energy consumption within buildings."

Line 345 Section 3.5. This section suddenly starts using personal pronouns (we) should be revised to be consistent with the rest of the paper.  AGREED AND NOW MOST "WE" ARE NOW REMOVED, THOUGH THEY DO OCCUR THROUGHOUT THE PAPER AT RARE INTERVALS.

A PROFESSIONAL ENGLISH EDITOR READ THE MS AT REVISION

Reviewer 2 Report

In my opinion, the paper shows a collection of air change data for different situations and you make a statistical analysis with these data. In order to improve its quality and the scientific soundness, you should add more information. I think it is necessary to add the adequate values of air change according to the standards and legislation and how these values could be achieved (that means possible solutions to improve the air change). Also, you say it is very important the energy efficiency. I suppose this energy efficiency is referred to the heating and cooling system. In order to improve the energy efficiency there are recovery heat units maintaining the ventilation level and reducing the level of pollutants but, without losing energy. The installation of these units is possible in some cases. These systems could be a good solution.

After this recommendation, I have some questions/doubts:

  • Table 1 should also show values for air change in all these cases, not only the method for the calculation.
  • In Section 2, when you explain the materials and methods, you should show the experimental equipment used in this research.
  • In line 153, you write units m h-1, is it correct? Or it will be m h-3?
  • Figure 2 shows the data for different type of buildings. In Hong Kong, have you considered houses? In this Figure, this data does not appear. Please check it.  
  • In section 3.2, please can you explain how you obtain the value of n?
  • In Line 320, I think there is an error. Please check it.

And finally, the English must be revised, there are several mistakes throughout the paper.

Author Response

REFEREE 2

Comments and Suggestions for Authors

In my opinion, the paper shows a collection of air change data for different situations and you make a statistical analysis with these data. In order to improve its quality and the scientific soundness, you should add more information. I think it is necessary to add the adequate values of air change according to the standards and legislation and how these values could be achieved (that means possible solutions to improve the air change). Also, you say it is very important the energy efficiency. I suppose this energy efficiency is referred to the heating and cooling system. In order to improve the energy efficiency there are recovery heat units maintaining the ventilation level and reducing the level of pollutants but, without losing energy. The installation of these units is possible in some cases. These systems could be a good solution.

AGREED

EXPANDED OUR CONCLUSION BY ADDING: “Energy efficiency in buildings is important to Hong Kong as air conditioning accounted for some 31% of the regions energy use according to the Hong Kong Energy End-use Data 2019, published by its Electrical and Mechanical Services Department. There have long been pressures to reduce energy consumption with air tightness as a key approach. It may be possible to improve the energy efficiency through heat recovery units while maintaining the present ventilation level, but these have not been widely applied. Future work needs to look at the role played by wind and stack effects in air exchange in high-rise apartments. Additionally, there needs to be more work on the way air moves between common areas and other apartments of tall buildings, as currently the balance of air deriving for outside and indoor spaces is not well understood.”

REGULATIONS NOW ADDED IN TWO POINTS CLOSE WHERE WE EXAMINE GOOD PRACTICE:

1/ THE MEASURED VALUES CAN BE COMPARED WITH THE STANDARDS TABULATED BY PAN [20] WHICH ARE TYPICALLY 2.8-6 H-1 FOR MOST EUROPEAN COUNTRIES, ALTHOUGH FINLAND WITH HARSH WINTERS HAS SET 1 H-1.

2/ THE MEASURED VALUES CAN ALSO BE COMPARED WITH GUIDELINE VALUES OF 3 M H-1 FOR ESTONIA AND FOR 1 M H-1 CANADA AS TABULATED BY PAN [20].

After this recommendation, I have some questions/doubts:

Table 1 should also show values for air change in all these cases, not only the method for the calculation.    THESE ARE ADDED TO THE TABLE NOW

In Section 2, when you explain the materials and methods, you should show the experimental equipment used in this research.

NOW MENTION THAT “HERE WE USED A RETROTEC BLOWER DOOR SYSTEM (RETROTEC, EVERSON, WA USA)… “

In line 153, you write units m h-1, is it correct? Or it will be m h-3?  I SEE WE HAVE CREATED CONFUSION SO THE SENTENCE HAS BEEN REWRITTEN - "WHICH CAN BE CONSIDERED THE FLOW THROUGH A GIVEN AREA OF THE BUILDING ENVELOPE (I.E. A FLUX) AND IN THE CASE OF A VOLUME OF AIR MOVING INTO A BUILDING HAS THE UNITS: m h-1 (ALTHOUGH OFTEN WRITTEN m3 h-1/m2), SO CAN BE THOUGHT OF AS A PERMEATION VELOCITY."

Figure 2 shows the data for different type of buildings. In Hong Kong, have you considered houses? In this Figure, this data does not appear. Please check it.   THERE ARE NO MEASUREMENTS FOR HOUSES IN HONG KONG - THEY ARE INCREDIBLY RARE AND USUALLY ONLY FOUND IN THE COUNTRYSIDE, SO THIS IS NOW NOTED IN THE TEXT "HOUSES ARE VERY RARE IN HONG KONG AND ARE TYPICALLY IN RURAL AREAS, SO HAVE NOT BEEN STUDIED." 

In section 3.2, please can you explain how you obtain the value of n? EXPLAINED IN THE TEXT AS - "THE SOFTWARE ASSOCIATED WITH RETROTEC 6100 BLOWER DOOR SYSTEM CALCULATES THIS VIA LEAST SQUARES."

In Line 320, I think there is an error. Please check it. YES DELETED

And finally, the English must be revised, there are several mistakes throughout the paper.

A PROFESSIONAL ENGLISH EDITOR READ THE MS AT REVISION

Reviewer 3 Report

Review of the paper urbansci-782840 “Air Change in Low and High-Rise Apartments” by Lai, Ridley, and Brimblecombe.

The paper is focused on the measurement of the ventilation in dwelling, both high-rise and low-rise, in buldings in Hong Kong. A discussion/comparison with other measurements performed worldwide is also provided. The paper is interesting for the different measurement methodologies used and the large literature comparison with similar studies.

Nonetheless some aspects should be improved:

  • The authors named the air exchange rate obtained from the Blower Door Test as “air exchange rate, k”. This is misleading since the air exchange rate (ACH) obtained from BDT is the result of tests performed at an indoor-outdoor pressure difference at 50 Pa! I strongly suggest to change the terms and symbols indicating the air exchange rate for BDT and CO2 decay tests. The latter is really an air exchange rate under typical indoor-outdoor pressure difference. Maybe, the authors could use k50 and k to differentiate the two terms and, moreover, they should avoid sentences like “apartments show a median of 7 air changes per hour” when they refer to air exchange rate from BDT.
  • Section 2.3. Following my previous comment, I strongly suggest the authors to make clear that the BDT is not meant to measure the air exchange rate under typical indoor-outdoor pressure differences.
  • Section 2.4. More details on the methodology adopted to perform the CO2 decay tests should be reported. As an example, it is not clear how the CO2 was injected in the dwellings (CO2 tanks? CO2 emitted by people?). Moreover, the authors should state here (not just in the results) that the three CO2 decay measurements were performed for each dwelling. The authors should clearly state that this is due to the fact that higher variations of the air exchange rates are expected as a function of the outdoor meteoclimatic conditions (T and wind). This is even more important in tall buildings where higher wind velocities are expected with respect to low-rise ones.
  • I would have expected more interesting discussions on the results. The current version of the results is more appropriate for a technical report (or a local study) than a scientific paper. Indeed, the current version of the discussion of the results is mainly based on the comparison with other buildings worldwide. Maybe further analyses could be carried out to improve the novelty of the paper and increase its scientific soundness. As an example, how the air exchange rates of the BDT and CO2 decay tests are correlated? Is it possible to consider a constant correction factor (as the semi-empirical correction factor obtained by the American researchers that you cited in the paper)? Is there an effect of the level (height) of the dwelling on the air exchange rate measured through the CO2 decay tests? …and the wind intensity and velocity?
  • The lack of proper discussion (besides the comparison with other situations worldwide) clearly comes to light in the abstract and conclusions. Indeed, in order to enhance them, the authors have had to include some speculations that go beyond what they really obtained from their work (e.g. in the abstract “Nevertheless, a lack of ventilation exposes residents of tiny apartments to cooking fumes and CO2. Establishing the right level of ventilation in buildings is increasingly important in an urbanising world”...this is not a finding)
  • A check of the English language is suggested.

Minor comments:

  • 1 the references to sites 8-9-10 are missing
  • Table 2. It is not clear which site belongs to the different building (e.g. Sites 1, 2 ad 3 are located in High-Rise 34?). Please modify the table. Please harmonize the figures used to express the temperature values (26.5 °C should be 27 °C).
  • Raw 122 – The Authors should refer to the international standard ISO 9972:2015
  • Raw 166 – The  Authors should refer to the international standard ISO 20485:2017
  • Raw 238 - The authors should better explain this sentence: ”These differences in response under depressurization and pressurization can suggest different approaches to maintenance of the inner and outer wall layers of the buildings.” and in general also the causes between these different behaviors in pressure and depression mode.

Author Response

REFEREE 3

Review of the paper urbansci-782840 “Air Change in Low and High-Rise Apartments” by Lai, Ridley, and Brimblecombe.

 The paper is focused on the measurement of the ventilation in dwelling, both high-rise and low-rise, in buldings in Hong Kong. A discussion/comparison with other measurements performed worldwide is also provided. The paper is interesting for the different measurement methodologies used and the large literature comparison with similar studies.

Nonetheless some aspects should be improved:

INDEED THANKS FOR THESE YOUR POINTS FORCED US TO REALLY THINK ON A NUMBER OF IMPORTANT ISSUES

The authors named the air exchange rate obtained from the Blower Door Test as “air exchange rate, k”. This is misleading since the air exchange rate (ACH) obtained from BDT is the result of tests performed at an indoor-outdoor pressure difference at 50 Pa! I strongly suggest to change the terms and symbols indicating the air exchange rate for BDT and CO2 decay tests. The latter is really an air exchange rate under typical indoor-outdoor pressure difference. Maybe, the authors could use k50 and k to differentiate the two terms and, moreover, they should avoid sentences like “apartments show a median of 7 air changes per hour” when they refer to air exchange rate from BDT.

Section 2.3. Following my previous comment, I strongly suggest the authors to make clear that the BDT is not meant to measure the air exchange rate under typical indoor-outdoor pressure differences.

GOOD POINTS - WE ARE NOW MORE CAREFULLY AND TRY TO REFER TO ASSOCIATE BLOWER DOOR TESTS WITH THE 50 Pa DIFFERENCE.  WE HAVE LABELLED THESE k50 where it is measured at 50 Pa  as suggested  and kACH for the parameter more generally (i.e. at a range of pressures) and kCO2 for the CO2 measurements (indeed we adopted that subscript for eq 5) .  

Section 2.4. More details on the methodology adopted to perform the CO2 decay tests should be reported. As an example, it is not clear how the CO2 was injected in the dwellings (CO2 tanks? CO2 emitted by people?). ADDED: SOME INCENSE STICKS WERE BURNT TO RAISE CO2 TO EXCEED 1000 PPM, THE SOURCE WAS THEN REMOVED

 Moreover, the authors should state here (not just in the results) that the three CO2 decay measurements were performed for each dwelling. The authors should clearly state that this is due to the fact that higher variations of the air exchange rates are expected as a function of the outdoor meteoclimatic conditions (T and wind). This is even more important in tall buildings where higher wind velocities are expected with respect to low-rise ones.

INDEED CORRECT SO WE HAVE NOW ADDED MORE: “The measurements were made at midnight, a time when external sources were at a minimum, so ambient CO2 concentrations are almost constant and wind speeds low, such that the effects of wind on air change were kept to a minimum, as these can cause variability in tall buildings.”

I would have expected more interesting discussions on the results. The current version of the results is more appropriate for a technical report (or a local study) than a scientific paper. Indeed, the current version of the discussion of the results is mainly based on the comparison with other buildings worldwide. Maybe further analyses could be carried out to improve the novelty of the paper and increase its scientific soundness. As an example, how the air exchange rates of the BDT and CO2 decay tests are correlated? Is it possible to consider a constant correction factor (as the semi-empirical correction factor obtained by the American researchers that you cited in the paper)?

THIS IS AN EXCELLENT POINT AND HAS BEEN DEALT WITH IN DETAIL AND WE HAVE INTRODUCED A NEW FIGURE TO COVER IT.  “Figure 4. (a) Comparison between the air change rate measured with the CO2 tracer and the blower-door at 50 Pa. The line represents the 20:1 slope expected for US houses [24]. (b) Comparison between the air change rate measured with the CO2 tracer and blower-door measurements at 50 Pa extrapolated to the natural pressure difference of each apartment (eq. 1). The line represents the 1:1 slope for perfect agreement. Note high-rise apartments: dark triangles; low-rise apartments: squares.” 

IT IS DISCUSSED IN A NEW PARAGRAPH “Typically, measurements of air change are made at 50 Pa, but these can be difficult to translate to the natural air change rate measured using a tracer, such as the CO2 used here. Sherman and his co-workers [24] suggest that the air change rate under natural conditions estimated with a tracer such as CO2 is typically k50/20, the factor of 20 is derived from fits to a large amount of data from houses in the US. In the current study the values for this factor ranged between 13 and 30, as suggested in Figure 4(a), by the 1:20 line. The high variability shown in this factor was not surprising given the complexities of stack and wind effects in tall buildings; besides this our sample size is small (N=6). Equation 1 allows blower-door measurements of air change at higher pressures to be extrapolated to the natural pressure differences measured in the study apartments (listed in Table 2). These are displayed in Figure 4(b) and while the calculated values of air change rate are a little higher than those measured with the CO2 tracer they are reasonably close to the 1:1 line.”

Is there an effect of the level (height) of the dwelling on the air exchange rate measured through the CO2 decay tests? …and the wind intensity and velocity?

THIS HAS BEEN STUDIED IN HIGH RISE BUILDINGS IN KOREA, SO IS NOW REFERRED TO IN SECTION 3.5 AND ALSO IN THE CONCLUSION AS A POTENTIAL FOR FUTURE STUDY

NOW ADDED WITHIN A MUCH CHANGE PARAGRAPH AT THE BEGINNING OF SECTION 3.5 “Stack effects are especially noticeable in high-rise buildings and cause elevator doors to stick, make it difficult to open doors to apartments and increase noise due to air leakage [52]. Typically tall buildings can also create problems for the blower-door method, due to changes in pressure from the operation of elevators or large shifts in external wind speed.  Our CO2 tracer experiments had to be conducted around midnight, to gain advantage from low wind speeds and elevator use, but this made them more likely to be stable. During the current blower-door measurements, the issue of wind did not prove a problem when using a pressure difference of 50 Pa, but care was taken to establish a stable baseline pressure calibration from the start. Following this, the blower-door tests were only conducted while stability could be maintained, and subsequent measurements repeated to reach an acceptable level of certainty.”

The lack of proper discussion (besides the comparison with other situations worldwide) clearly comes to light in the abstract and conclusions. Indeed, in order to enhance them, the authors have had to include some speculations that go beyond what they really obtained from their work (e.g. in the abstract “Nevertheless, a lack of ventilation exposes residents of tiny apartments to cooking fumes and CO2. Establishing the right level of ventilation in buildings is increasingly important in an urbanising world”...this is not a finding)

AGREE SO WE HAVE EXPANDED THE SENSE OF SCIENTIFIC NOVELTY OF FINDINGS IN TERMS OF THE COMPARISON OF CO2 AND EXTRAPOLATED VENTILLATION RATE IN SECTION 3.3 AND 3.5 AS MENTIONED ABOVE. WE HAVE REMOVED THE DISCUSSION ON   " lack of ventilation exposes residents of tiny apartments " FROM THE ABSTRACT - WE AGREE THIS IS NOT APPROPRIATE

WE AGREE THIS IS NOT A FINDING - IT IS MORE OF A POLITICAL COMMENT ABOUT THE PRESSURE TO LOWER VENTILATION RATES.  IT ALSO ATTEMPTS TO ALIGN THE RESEARCH WITH A GENERAL NOTION OF "URBAN SCIENCE". ANOTHER REFEREE ASKED FOR MORE DISCUSSION OF ISSUES SUCH AS SOLUTION AS AND HEAT EXCHANGERS SO THESE WERE ALSO ADDED TO THE CONCLUSION.

WHILE IT IS TRUE CONCLUSIONS SHOULD DERIVE FROM THE WORK IT ALSO SEEMS THAT CONCLUSIONS ARE THE PLACE WHERE AUTHORS MIGHT LEGITIMATELY INDULGE IN BROADER THOUGHTS ABOUT THE POSSIBLE IMPACT OF THEIR RESEARCH. WE HAVE TRIED TO CHANGE THIS SUCH THAT IT IS CLEAR THAT THESE VIEWS ARE NOT FINDINGS. “Hong Kong sometimes experiences poor air quality on its busy streets, and aloft there is the potential for exposure to air pollutants transported down to the Pearl River Delta, from an increasingly industrialized Guangdong Province. Air-tight buildings can prevent the invasion of polluted air during air pollution episodes. At the other end of the scale it is apparent that poorer people in Hong Kong living in tiny dwellings suffer from insufficient ventilation. In these dwellings low air exchange may inhibit the removal of cooking fumes and carbon dioxide. The competing demands experienced in our cities is nicely reflected in the problem of ventilation described here: there are difficulties in establishing the right level of ventilation in the buildings of an increasingly urbanized world.”

A check of the English language is suggested. A PROFESSIONAL ENGLISH EDITOR READ THE MS AT REVISION

Minor comments:

1 the references to sites 8-9-10 are missing - LABELLED AND NOTED THAT THIS BUILDING HAS NO PHOTOGRAPH

Table 2. It is not clear which site belongs to the different building (e.g. Sites 1, 2 ad 3 are located in High-Rise 34?). Please modify the table. Please harmonize the figures used to express the temperature values (26.5 °C should be 27 °C). Changed 26.5 and put in lines to clarify sites  CHANGED 26.5 AND PUT IN LINES IN TABLE TO CLARIFY SITES

Raw 122 – The Authors should refer to the international standard ISO 9972:2015 NOW INSERTED INTO TEXT

Raw 166 – The  Authors should refer to the international standard ISO 20485:2017  NOW INSERTED INTO TEXT

Raw 238 - The authors should better explain this sentence: ”These differences in response under depressurization and pressurization can suggest different approaches to maintenance of the inner and outer wall layers of the buildings.” and in general also the causes between these different behaviors in pressure and depression mode. NOW READS "THESE DIFFERENCES IN RESPONSE UNDER DEPRESSURIZATION AND PRESSURIZATION CAN SUGGEST DIFFERENT APPROACHES TO CONSTRUCTION OR MAINTENANCE OF THE INTEGRITY OF INNER AND OUTER WALL LAYERS OF THE BUILDINGS.  UNDER PRESSURIZATION GREATER INDOOR PRESSURE MEANS THAT THE INNER WALL LAYER WILL BE PUSHED OUTWARD SO SOME POINTS MAY LEAK MORE, SUCH AS FAN LOUVRES; WHILE DURING DEPRESSURIZATION MEANS AIR IS SUCKED INSIDE SO SOME OUTER WALL LAYERS MAY BEND INWARDS AND LEAK.  "

Round 2

Reviewer 2 Report

Thank you for your answers. Now, the data you show in this paper are clearer.

Reviewer 3 Report

The requested corrections have been implemented and the paper appears to have very improved. Then the paper is accepted in present form.